

# Factors associated with elevated blood pressure or hypertension in Afro-Caribbean youth: a cross-sectional study

Trevor S. Ferguson[1], Novie O.M. Younger-Coleman[1], Marshall K. Tulloch-Reid[1], Nadia R. Bennett[1], Amanda E. Rousseau[1], Jennifer M. Knight-Madden[1], Maureen E. Samms-Vaughan[2], Deanna E. Ashley[3] and Rainford J. Wilks[1]

[1] Caribbean Institute for Health Research, University of the West Indies, Mona, Kingston, Jamaica
[2] Department of Child Health, University of the West Indies, Mona, Kingston, Jamaica
[3] School of Graduate Studies and Research, University of the West Indies, Mona, Kingston, Jamaica

## ABSTRACT

**Background.** Although several studies have identified risk factors for high blood pressure (BP), data from Afro-Caribbean populations are limited. Additionally, less is known about how putative risk factors operate in young adults and how social factors influence the risk of high BP. In this study, we estimated the relative risk for elevated BP or hypertension (EBP/HTN), defined as BP ≥ 120/80 mmHg, among young adults with putative cardiovascular disease (CVD) risk factors in Jamaica and evaluated whether relative risks differed by sex.

**Methods.** Data from 898 young adults, 18–20 years old, were analysed. BP was measured with a mercury sphygmomanometer after participants had been seated for 5 min. Anthropometric measurements were obtained, and glucose, lipids and insulin measured from a fasting venous blood sample. Data on socioeconomic status (SES) were obtained via questionnaire. CVD risk factor status was defined using standard cut-points or the upper quintile of the distribution where the numbers meeting standard cut-points were small. Relative risks were estimated using odds ratios (OR) from logistic regression models.

**Results.** Prevalence of EBP/HTN was 30% among males and 13% among females ($p < 0.001$ for sex difference). There was evidence for sex interaction in the relationship between EBP/HTN and some of risk factors (obesity and household possessions), therefore we report sex-specific analyses. In multivariable logistic regression models, factors independently associated with EBP/HTN among men were obesity (OR 8.48, 95% CI [2.64–27.2], $p < 0.001$), and high glucose (OR 2.01, CI [1.20–3.37], $p = 0.008$), while high HOMA-IR did not achieve statistical significance (OR 2.08, CI [0.94–4.58], $p = 0.069$). In similar models for women, high triglycerides (OR 1.98, CI [1.03–3.81], $p = 0.040$) and high HOMA-IR (OR 2.07, CI [1.03–4.12], $p = 0.039$) were positively associated with EBP/HTN. Lower SES was also associated with higher odds for EBP/HTN (OR 4.63, CI [1.31–16.4], $p = 0.017$, for moderate vs. high household possessions; OR 2.61, CI [0.70–9.77], $p = 0.154$ for low vs. high household possessions). Alcohol consumption was associated with lower odds of EBP/HTN among females only; OR 0.41 (CI [0.18–0.90], $p = 0.026$) for drinking <1 time per week vs. never drinkers, and OR 0.28 (CI [0.11–0.76], $p = 0.012$) for drinking ≥3 times per week vs. never drinkers. Physical activity was inversely associated with EBP/HTN in both males and females.

Corresponding author
Trevor S. Ferguson,
trevor.ferguson02@uwimona.edu.jm,
trevor.ferguson02@gmail.com

**Conclusion**. Factors associated with EBP/HTN among Jamaican young adults include obesity, high glucose, high triglycerides and high HOMA-IR, with some significant differences by sex. Among women lower SES was positively associated with EBP/HTN, while moderate alcohol consumption was associated lower odds of EBP/HTN.

## INTRODUCTION

High blood pressure (BP) is the leading risk factor for the global burden of disease, accounting for approximately 7% of global disability adjusted life years (*Lim et al., 2012*). Recent studies suggest that while the prevalence of hypertension is decreasing in high-income countries, prevalence is increasing in low and middle-income countries, with the largest increase seen in countries in sub-Saharan Africa (*Mills et al., 2016*; *NCD Risk Factor Collaboration, 2016*). The adverse effect of high BP, particularly increased risk of coronary heart disease and stroke, is continuous and graded throughout the range of systolic blood pressure (SBP) and diastolic blood pressure (DBP), down to levels of 115 mmHg and 75 mmHg, respectively (*Lewington et al., 2002*). Additionally, it has been estimated that approximately 50% of disease burden attributable to high BP occurs at levels below the 140/90 mmHg cut-off point traditionally used to define hypertension (*Poulter, Prabhakaran & Caulfield, 2015*). Recently the American College of Cardiology and American Heart Association (ACC/AHA) proposed new guidelines for the evaluation and management of high BP (*Whelton et al., 2017*). In this guideline, normal BP is defined having SBP <120 mmHg and DBP <80 mmHg; elevated BP is defined as SBP of 120-129 mmHg and DBP <80 mmHg; and hypertension defined as SBP ≥130 mmHg or DBP >80 mmHg. However, most of the available data on prevalence of hypertension have used the criteria from the Seventh Report of the Joint National Committee on the Prevention, Detection, Evaluation and Treatment of High Blood Pressure (JNC 7), where hypertension is defined as SBP ≥140 mmHg or DBP ≥90 mmHg, and SBP of 120-139 mmHg or DBP of 80-89 is classified as prehypertension (*Chobanian et al., 2003*). Studies reporting prevalence estimates for children or adolescents <18 years old often use criteria from The Fourth Report on the Diagnosis, Evaluation, and Treatment of High Blood Pressure in Children and Adolescents by the National High Blood Pressure Education Program (NHBPEP) (*National High Blood Pressure Education Program Working Group on High Blood Pressure in Children and Adolescents, 2004*).

Reported prevalence of hypertension, using JNC 7 or NHBPEP criteria, in adolescents and young adults vary widely, with estimates generally ranging from about 2% among 15–34 year-olds in Italy up to 19% among young adults 24–34 years old in the USA (*Battistoni et al., 2015*). However, prehypertension appears to be common in adolescents and young adults, with prevalence estimates ranging from 12%–45% in various studies from countries such as India, Uganda, United States and Jamaica (*Amma, Vasudevan & Akshayakumar,*

*2015*; *Ferguson et al., 2011b*; *Kayima et al., 2015*; *Kini et al., 2016*; *Redwine & Daniels, 2012*). Given that high BP in childhood has been shown to track into adulthood (*Bao et al., 1995*; *Chen & Wang, 2008*), studies of high BP in youth provide essential information to inform interventions that would reduce the adverse effects of high BP on cardiovascular health.

The aetiology of hypertension is multi-factorial, with complex interactions between genetic, environmental, behavioural and social factors (*Lloyd-Jones & Levy, 2013*; *Poulter, Prabhakaran & Caulfield, 2015*; *Victor, 2015*). Established risk factors for hypertension include increasing age, higher levels of adiposity, high dietary sodium, high alcohol consumption, family history of hypertension and lower socioeconomic status (*Lloyd-Jones & Levy, 2013*). Underlying mechanism include activation of the sympathetic nervous system, disorders of the renin-angiotensin aldosterone pathways, disorders of renal regulation of sodium balance, insulin resistance, inflammation, arterial stiffness and foetal programming (*Acelajado, Calhoun & Oparil, 2013*; *Victor, 2015*).

Complications of hypertension vary with race/ethnicity and it is conceivable that the mechanisms underlying both aetiology and complications could vary similarly (*Jones & Hall, 2006*; *Lackland, 2014*). Additionally, less is known about how these factors operate in young African origin populations and how social factors, particularly in a developing country context, influence the risk of high BP.

In Jamaica, the prevalence of hypertension (using the JNC 7 criteria) among persons 15–74 years old was estimated at 20% in 2001, and 25% in 2008 (*Ferguson et al., 2011a*). The prevalence of prehypertension was 30% in 2001 and 35% in 2008, and was shown to be associated with other cardiovascular disease (CVD) risk factors and high rates of progression to hypertension (*Ferguson et al., 2011a*; *Ferguson et al., 2010c*; *Ferguson et al., 2008*). Among 15–19 year-old youth, the prevalence of prehypertension in 2006 was 29% (*Ferguson et al., 2011b*). More recently, the Modeling the Epidemiological Transition Study reported prevalence of hypertension among urban Jamaicans 25–45 years old, with prevalence estimates of 6.8% among men and 10% among women (*Cooper et al., 2015*). This study also found that the prevalence of CVD risk factors was not always consistent with that expected, with Jamaican women having lower diabetes prevalence despite high obesity prevalence and South African men having higher prevalence of hypertension despite lower adiposity (*Dugas et al., 2017*). Given the high burden of hypertension and prehypertension in Jamaica, studies evaluating the relative contribution of various risk factors would provide necessary information to direct public health interventions. This paper therefore evaluates the association between putative risk factors and elevated BP or hypertension (EBP/HTN), defined as BP $\geq$120/80 mmHg, among Afro-Caribbean youth. Specifically, we aimed to estimate the relative risk for having EBP/HTN among participants with putative CVD risk factors, and to evaluate whether there were significant sex differences in risk factors for EBP/HTN.

## METHODS

### Data sources

We conducted a cross-sectional analysis using data from the third follow up of the Jamaica 1986 Birth Cohort Study (*Ferguson et al., 2010a*; *McCaw-Binns et al., 2011*). This study

is a longitudinal study of persons, born in Jamaica in September and October of 1986, and who were a part of the Jamaica Perinatal Mortality Survey (*Ashley, McCaw-Binns & Foster-Williams, 1988*). Details on this cohort have been previously published (*Bennett et al., 2014*; *McCaw-Binns et al., 2011*). For this analysis, we used data from 409 males and 489 females, 18–20 years old, collected in the third follow up of the cohort between March 2005 and February 2007. The study was approved by the University of the West Indies/Faculty of Medical Sciences Ethics Committee. Participants provided written informed consent prior to measurements being done.

## Measurements and definitions

All data collection and measurements were done by trained research nurses. We obtained data on demographic characteristics, general health, medical history, behavioural health risk factors and socioeconomic status via questionnaire. Additionally, we obtained anthropometric and BP measurements and performed venepuncture for analysis of blood glucose, lipids, insulin and creatinine. A timed urine sample was obtained for measurement of urinary albumin excretion.

BP was measured with a mercury sphygmomanometer after the participant had been seated for 5 min. BP measurement followed a standardized protocol developed for the International Collaborative Study of Hypertension in Blacks (*Ataman et al., 1996*). Three BP measurements were taken at 1-minute intervals, with the mean of the second and third measurements being used for analysis. EBP/HTN was defined as SBP $\geq$120 mmHg or DBP of $\geq$80 mmHg, corresponding to the prehypertension and hypertension categories of JNC 7 and the elevated BP and hypertension categories of the 2017 ACC/AHA guidelines (*Chobanian et al., 2003*; *Whelton et al., 2017*). None of the participants were on medication for elevated blood pressure at the time of assessment.

Weight was measured using a portable digital scale, which was calibrated daily. Height was measured using a portable stadiometer. Waist and hip circumference were measured using a non-stretchable nylon tape measure. Body mass index (BMI) was calculated as weight in kilograms divided by the square of height in metres and BMI categories defined using the World Health Organization categories: underweight (BMI <18.5 kg/m$^2$), normal weight (BMI 18.5–24.9 kg/m$^2$), overweight (BMI 25.0–29.9 kg/m$^2$), obese (BMI $\geq$30 kg/m$^2$) (*World Health Organization, 1995*). The normal weight category was used as the reference group. Central obesity was defined as a waist circumference $\geq$80 cm for women and $\geq$94 cm for men as recommended for African Origin populations in the 2009 Consensus Criteria for the Metabolic Syndrome (*Alberti et al., 2009*). Waist-to-hip ratio was calculated by dividing waist circumference by hip circumference. High waist-to-hip ratio was defined using cut-points recommended by *Lean, Han & Morrison (1995)* as $\geq$0.95 for males and $\geq$0.80 for females.

Venous blood was collected after an overnight fast of at least eight hours. Samples were analysed using standard laboratory protocols for measurement of fasting glucose, lipids, fasting insulin and serum creatinine. White blood cell count and high sensitivity C-reactive protein (hsCRP) were measured as markers of inflammation.

Details of laboratory procedures have been previously published (*Bennett et al., 2014*; *Ferguson et al., 2010a*; *Rocke et al., 2015*; *Tulloch-Reid et al., 2010*). In brief, glucose was measured using the glucose oxidase method (Alcyon, Analyzer); total cholesterol, triglycerides, and high density lipoprotein cholesterol (HDL) were measured directly using enzymatic methods (Abbott Spectrum Analyzer), while low density lipoprotein cholesterol (LDL) was calculated using the Friedewald equation (total cholesterol−HDL−[TG/2.18], with all concentrations given in mmol/L). Serum creatinine was measured using Jaffe's reaction on the Alcyon 300 Chemistry Analyser (Abbott, Chicago, IL, USA), while fasting insulin was measured using a chemiluminescent immunoassay (IMMULITE; Diagnostic Products Corporation, Los Angeles, CA, USA); hsCRP was measured using an IMMULITE immunoassay (Siemens Medical Solution Diagnostics, Los Angeles, CA). Microalbumin was measured from a timed urine specimen using a chemiluminescent immunoassay method using the IMMULITE Immunoassay System (Siemens, Los Angeles, CA).

Elevated glucose (≥5.6 mmol/L), elevated triglycerides (≥1.7 mmol/L) and low levels of high density lipoprotein cholesterol [HDL] (<1.0 mmol/L for males and <1.3 mmol/L for females) were defined using the metabolic syndrome criteria (*Alberti et al., 2009*). High total cholesterol (≥5.2 mmol/L) and high levels of low density lipoprotein [LDL] cholesterol (≥4.1 mmol/L) were defined using the National Cholesterol Education Program Adult Treatment Panel III (ATP III) criteria (*Expert Panel on Detection Evaluation and Treatment of High Blood Cholesterol in Adults, 2001*). For analyses including glucose and triglycerides values in the upper quintile of the distribution were defined as elevated, because the proportion of participants meeting the metabolic syndrome cut-points was very small, thus resulting in too few participants for multivariable analyses. High hsCRP was defined as >3.0 mg/L, with values >10 mg/L set to missing, as recommended by the American Heart Association and Centers for Disease Control (*Pearson et al., 2003*). Insulin resistance was estimated using the Homeostasis Model Assessment (HOMA-IR) equations (*Matthews et al., 1985*). Values for HOMA-IR were log transformed to account for non-normal distribution. Elevated HOMA-IR was classified as being in the upper quintile of the log-HOMA-IR. For this paper we chose to dichotomize these characteristics in order to quantify the effect of being in an abnormal (high risk) category and facilitate the tailoring of public health messages aimed at risk reduction. Urine albumin and creatinine levels were used to calculate the albumin to creatinine ratio (ACR) and elevated urine albumin defined as ACR ≥30 mg/g as recommended by the 2012 Kidney Disease Improving Global Outcomes (KDIGO) Guidelines (*Kidney Disease: Improving Global Outcomes (KDIGO) CKD Work Group, 2013*).

Socioeconomic status was assessed using data collected using a locally developed questionnaire on parental education and occupation, and number of household possessions. The specific questionnaire items are included in the supplementary files available online. Data on education was collected as the highest level of education attained by either parent or guardian and then categorized as: post-secondary, secondary, or less than secondary. In Jamaica, children are required to complete a mandatory six years of elementary school (Grades 1–6) and five years of high school (Grades 7–11 or first to fifth form), after which they graduate from high school. Children may spend an extra

two years in sixth form (Grades 12 and 13) before going on to university or college. In this classification, post-secondary education includes persons who completed college or university and persons with vocational training obtained after completing high school. Secondary education indicates persons who completed high school (up to grade 11) and less than secondary includes persons who did not complete high school (i.e., high school education up to grade 10 or below). Occupational categories were defined using the occupation of the household head and coded using the Jamaica Standard Occupational Classification (JSOC) (*Statistical Institute of Jamaica, 1995*). For this report occupation categories were classified as professionals or managers, office, service or trade workers, and semi-skilled or unskilled workers. For household possessions, participants were asked to indicate whether they had items from a list of 17 household possessions and given one point for each item. They were then classified in three possession score categories based on the distribution of items: low (0–9 items), moderate (10–14 items), and high (15–17 items). The list of items included in the possession score is shown Table S1. Cut-points for this classification was chosen based on the finding that the majority of participants had 10–14 items, so that the groups with 0–9 items would represent the lower end of the distribution and 15–17 items the upper end of the distribution.

Data on physical activity, smoking (cigarettes or marijuana) and alcohol consumption were also collected via questionnaire. For smoking, participants were classified as current smokers or non-smokers, while for alcohol consumption participants were classified as: 'never drank alcohol', 'rarely drinks alcohol' (<1 time per week), 'drinks alcohol 1–2 times per week', or 'drinks alcohol ≥3 times per week'. Physical activity was classified based on the time spent doing sports or exercise during leisure time using a locally developed questionnaire. Questionnaire items included questions on time spent doing active sports or other activities such as brisk walking, jogging, lifting weights, dance classes, and workout at a gym. The specific questionnaire items are included in the online supplementary files. Physical activity assessed using this questionnaire was shown to be more strongly associated with measures of obesity than the International Physical Activity Questionnaire (IPAQ) (*Younger et al., 2007*). Participants with no leisure time physical activity were classified as low physical activity level, those with <3.5 h per week as moderate physical activity level and those with 3.5 h or more per week as high physical activity level.

## Sample size and power

Given that we had a fixed available sample size of 409 males and 489 females and that we performed sex-specific analyses, we estimated the maximum detectable odds ratio instead of sample size. These estimates were computed for males and females, separately, using the power twoproportions command available in Stata 14 (*StataCorp, 2015c*). We estimated the prevalence of the outcome variable (EBP/HTN), from the sample at 30% for males and 13% for females, and used these and the available sample size to compute the maximum detectable odds ratio for exposures, with proportion exposed ranging from 0.1 to 0.5 for power of 80% at the 5% significance level. For males, the given sample size of 409 had 80% power to detect odds ratio of 1.84 if the proportion exposed was 0.5 and 2.13 if the

proportion exposed was 0.2. For females the given sample size of 489 had 80% power
to detect odds ratio of 2.02 if the proportion exposed was 0.5 and 2.34 if the proportion
exposed was 0.2.

## Statistical methods

We performed data analysis with Stata version 14 .1 software (Stata Corp., College Station,
TX, USA). We obtained descriptive statistics (means and proportions) for outcome and
explanatory variables within and across sex and blood pressure categories. If data were
highly skewed, we reported the median and interquartile range instead of the mean and
standard deviation. Proportions were compared using the Pearson's chi-squared test
or Fisher's exact tests, as appropriate. Differences in means were compared using the
unequal variance two sample $t$-test. Differences in medians were compared using the
non-parametric equality of median test available in Stata (*StataCorp, 2015a*).

Logistic regression and two-way analysis of variance (ANOVA) models were used to
determine if there was evidence for sex interaction in the relationship between BP and
some of the explanatory variables. Results for analyses assessing interaction using the
logistic regression models are shown in Table S2. There was evidence for sex interaction in
the relationship between EBP/HTN and some of risk factors (obesity, central obesity and
household possessions), therefore we report sex-specific results for regression analyses.

We used multiple imputation by chained equations to account for missing data for some
explanatory variables. The proportion of complete cases, i.e., participants with no missing
values for any of the variables of interest, was 43% ($n = 384$); The majority of incomplete
cases (30% of participants) had only had one missing value; 14% had two missing values,
7% had three missing values and 7% had more than three missing values. Details on
the number of missing values for each variable are shown in Table S3. A comparison of
the complete cases vs. the incomplete cases revealed only minor differences. Participants
with missing values were more likely to have albuminuria, fewer household possessions,
lower physical activity and lower alcohol consumption, but had no statistically significant
differences for any other characteristics. Given the proportion of participants with at least
one missing value and the observed differences between complete and incomplete cases,
multiple imputation was used to improve the power of the study and reduce bias that may
be seen in the complete case analysis (*Nguyen, Carlin & Lee, 2017*; *White, Royston & Wood,
2011*). A stacked multiple imputed data set, consisting of the original dataset and 25 data
sets with imputations for missing values, was created using Stata's mi suite of commands
(*StataCorp, 2015b*). We compared imputed variable values to the observed values to ensure
that the imputed values were plausible; these data are shown in Table S4.

Bivariate logistic regression was used to assess the association between EBP/HTN and
individual explanatory variables. These bivariate models were estimated using Stata's mi
suite of commands and estimates combined by the software using Rubin rules (*Marshall
et al., 2009*; *StataCorp, 2015b*). For use in model selection, we extracted the first of the 25
imputed data sets and performed regular binary logistic regression on the single-imputed
data, as recommended by Wood, White and Royston (*Wood, White & Royston, 2008*). We

used the backwards stepwise regression algorithm available in Stata to identify variables for inclusion in the final model. All variables hypothesized to be associated with the outcome were included in the first multivariable model and $p$-value $> 0.2$ was used to remove variables from the model. We then used the Pearson and Hosmer-Lemeshow tests for goodness-of-fit to assess the models. Finally, we used Akaike information criterion (AIC) to determine whether to include or exclude specific variables from the final models. Final multivariable models were then run on the multiple imputed data set with 25 imputations, using Stata's mi suite of commands and estimates combined by the software using Rubin rules (*Marshall et al., 2009*; *StataCorp, 2015b*). To assess the potential impact of the imputed values on the final conclusions we also re-ran the final models without imputations (i.e., complete case analysis); these results are shown in Table S5.

## RESULTS

Summary statistics for demographic and biomedical measurements are shown in Table 1. Mean age at the time of the study was 18.8 years (SD = 0.61), with no sex difference. Compared to females, males had higher mean weight, height, SBP, DBP, fasting glucose, triglycerides and creatinine, while females had higher mean total cholesterol, LDL and HDL cholesterol. Females also had higher median hsCRP, fasting insulin concentration and HOMA-IR. Comparisons of participants' characteristics by BP categories are shown in Table S6. Overall, participants with EBP/HTN tended to have higher mean values of CVD risk factors.

Proportions of participants with EBP/HTN and other CVD risk factors, expressed as categorical variables, are shown is Table 2. Overall prevalence of EBP/HTN was 21% and was twice as high in men compared to women (30% vs. 13%, $p < 0.001$). The prevalence of elevated BP (SBP 120–129 mmHg, DBP <80 mmHg) was 9% (13% among males and 5% among females, $p < 0.001$), while hypertension (BP $\geq$130/80 mmHg) prevalence was 12% (17% among males and 8% among females, $p < 0.001$). Prevalence of obesity was 8% (6% among males and 10% among females, $p = 0.008$). The majority of participants were from middle-income households, with the household head having completed secondary level education and working as office, service, or trade workers. Low physical activity level was reported by 34% of participants and high physical activity by 24%. There were significant sex differences in physical activity among males compared to females ($p < 0.001$) with 47% of females reporting low physical activity levels compared to 18% among males, while high physical activity was reported by 38% of males compared to 13% among females. Cigarette smoking was reported by 14% of males and 6% of females ($p < 0.001$), while 31% of males and 7% of females smoked marijuana ($p < 0.001$). Males also reported higher levels of moderate ($\geq$3 times/week) alcohol consumption (38% vs. 19%, $p < 0.001$). Similar analyses stratified by BP category are shown in Table S7. Significant difference by blood pressure category were seen for BMI categories, central obesity and waist-to-hip ratio among males, and for number of household possessions among females.

**Table 1  Mean or median values for participant characteristics and putative hypertension risk factors for males, females and both sexes.**

| Characteristic | Male n = 409 Mean ± SD | Female n = 489 Mean ± SD | Both sexes N = 898 Mean ± SD |
|---|---|---|---|
| Age (years) | 18.8 ± 0.59 | 18.8 ± 0.62 | 18.8 ± 0.61 |
| Weight (kg)[***] | 71.1 ± 14.2 | 62.4 ± 15.5 | 66.4 ± 15.5 |
| Height (cm)[***] | 176.8 ± 6.5 | 163.6 ± 6.1 | 169.6 ± 9.1 |
| Body mass index (kg/m$^2$) | 22.7 ± 4.3 | 23.3 ± 5.6 | 23.0 ± 5.0 |
| Systolic blood pressure (mmHg)[***] | 113.9 ± 10.4 | 107.4 ± 8.8 | 110.3 ± 10.1 |
| Diastolic blood pressure (mmHg)[***] | 69.2 ± 10.3 | 66.9 ± 9.2 | 67.9 ± 9.8 |
| Waist circumference (cm) | 75.2 ± 10.8 | 73.9 ± 12.1 | 74.5 ± 11.5 |
| Hip circumference (cm)[**] | 94.4 ± 8.9 | 96.5 ± 11.0 | 95.5 ± 10.2 |
| Waist-to-Hip ratio[***] | 0.80 ± 0.08 | 0.77 ± 0.14 | 0.78 ± 0.12 |
| White blood cell count (cells × 10$^9$/L)[***] | 5.3 ± 1.6 | 6.4 ± 2.0 | 5.9 ± 1.9 |
| Fasting glucose (mmol/L)[***] | 4.7 ± 0.6 | 4.4 ± 0.4 | 4.6 ± 0.5 |
| Total cholesterol (mmol/L)[***] | 4.1 ± 0.8 | 4.5 ± 0.9 | 4.3 ± 0.9 |
| HDL cholesterol (mmol/L)[***] | 1.1 ± 0.2 | 1.2 ± 0.3 | 1.2 (0.3) |
| LDL cholesterol (mmol/L)[***] | 2.7 ± 0.7 | 3.0 ± 0.8 | 2.9 ± 0.8 |
| Triglycerides (mmol/L)[*] | 0.60 ± 0.26 | 0.56 ± 0.26 | 0.58 ± 0.26 |
| Creatinine (μmol/L)[***] | 80.5 ± 16.0 | 56.9 ± 25.5 | 67.7 ± 24.7 |
| | Median (IQR) | Median (IQR) | Median (IQR) |
| Urinary albumin (mg/g)[*] | 3.9 (2.5, 7.4) | 4.9 (2.5, 10.7) | 4.1 (2.5, 9.2) |
| hsCRP (mg/L)[***] | 0.5 (0.3, 1.3) | 0.9 (0.3, 2.3) | 0.7 (0.3, 1.8) |
| Fasting insulin (pmol/L)[***] | 4.4 (2.7, 7.1) | 6.8 (4.1, 10.1) | 5.8 (3.3, 8.8) |
| HOMA-IR[***] | 0.6 (0.3, 0.9) | 0.9 (0.5, 1.3) | 0.7 (0.4, 1.1) |

**Notes.**
[*]$p < 0.05$.
[**]$p < 0.01$.
[***]$p < 0.001$.

SD, standard deviation; HDL, high density lipoprotein; LDL, low density lipoprotein; IQR, interquartile range (values correspond to the 25th and 75th centiles); hsCRP, high sensitivity C-reactive protein; HOMA-IR, Homeostasis Model Assessment—Insulin Resistance.

Differences in means were compared using the two-sample $t$ test with unequal variances, while the differences in medians were computed using the nonparametric equality-of-medians test.

The results from bivariate analyses yielding sex specific odds ratios for the relationship between correlates and putative risk factors for EBP/HTN are shown in Table 3. Factors associated with EBP/HTN among males in bivariate analyses were: age, obesity, central obesity, high glucose, high triglycerides and high HOMA-IR. Among females, significant correlates were age, height, high triglycerides, high HOMA-IR and number of household possessions. There were no significant associations for general or central obesity among females, and no significant associations for measures of inflammation (hsCRP and white blood cell count) or urine albumin excretion in either sex.

**Table 2  Proportion of participants in categories for blood pressure and other CVD risk factors for males, females and both sexes.**

| Characteristic | Male $n = 409$ % ($n$) | Female $n = 489$ % ($n$) | Both sexes $N = 898$ % ($n$) |
|---|---|---|---|
| Elevated BP or hypertension (BP $\geq$120/80 mmHg)[***] | 29.8 (122) | 13.4 (66) | 20.9 (188) |
| Elevated BP (SBP120–129 & DBP <80 mmHg)[***] | 13.2 (54) | 5.3 (26) | 8.9 (80) |
| Hypertension (BP $\geq$130/80)[***] | 16.6 (68) | 8.2 (40) | 12.0 (108) |
| Stage 1 hypertension (BP 130–139/80–89)[***] | 14.7 (60) | 7.4 (36) | 10.7 (96) |
| Stage 2 hypertension (BP $\geq$140/90) | 2.0 (8) | 0.8 (4) | 1.3 (12) |
| Albuminuria | 5.1 (20) | 8.3 (39) | 6.8 (59) |
| Body mass index categories[***] | | | |
| *Underweight* (<18.5 kg/m$^2$) | 6.1 (25) | 14.5 (71) | 10.7 (96) |
| *Normal weight* (18.5–24.9 kg/m$^2$) | 76.3 (308) | 55.2 (270) | 64.4 (578) |
| *Overweight* (25–29.9 kg/m$^2$) | 13.0 (53) | 19.8 (87) | 16.7 (150) |
| *Obese* ($\geq$30 kg/m$^2$) | 5.6 (23) | 10.4 (51) | 8.2 (74) |
| Central obesity[a,***] | 5.1 (21) | 24.4 (119) | 15.6 (140) |
| High waist-to-hip ratio[b,***] | 1.0 (4) | 20.3 (99) | 11.5 (103) |
| Highest Education of Parent/Guardian[c] | | | |
| *Post-secondary* | 26.2 (89) | 29.8 (131) | 28.2 (220) |
| *Secondary* | 61.8 (210) | 55.4 (243) | 58.2 (453) |
| *Less than secondary* | 12.1 (41) | 14.8 (65) | 13.6 (106) |
| Occupation of household head | | | |
| *Professionals/Managers* | 23.5 (88) | 24.7 (114) | 24.1 (202) |
| *Office, service or trade workers* | 49.3 (185) | 50.9 (235) | 50.2 (420) |
| *Semi-skilled/Unskilled workers* | 27.2 (102) | 24.5 (113) | 25.7 (215) |
| Number of household possession | | | |
| *High* (15–17 items) | 16.9 (69) | 13.7 (67) | 15.2 (136) |
| *Moderate* (10–14 items) | 56.9 (232) | 54.2 (265) | 55.4 (497) |
| *Low* (0–9 items) | 26.2 (107) | 32.1 (157) | 29.4 (264) |
| Physical activity level[***] | | | |
| *High* | 37.5 (153) | 13.1 (64) | 24.2 (217) |
| *Moderate* | 44.4 (181) | 39.5 (193) | 41.7 (374) |
| *Low* | 18.1 (74) | 47.4 (232) | 34.1 (306) |
| Current cigarette smoking[***] | 13.7 (56) | 6.1 (30) | 9.6 (86) |
| Current marijuana smoking[***] | 31.3 (127) | 7.2 (35) | 18.1 (162) |
| Alcohol consumption[***] | | | |
| *Never drank alcohol* | 6.4 (26) | 13.2 (64) | 10.1 (90) |
| *Rarely drinks alcohol* | 26.4 (107) | 45.2 (219) | 36.6 (326) |
| *Drinks alcohol 1–2 times/week* | 29.1 (118) | 22.3 (108) | 25.4 (226) |
| *Drinks alcohol $\geq$3 times/week* | 38.0 (154) | 19.4 (94) | 27.9 (248) |

**Notes.**
[*]$p < 0.05$.
[**]$p < 0.01$.
[***]$p < 0.001$.

BP, blood pressure; SBP, systolic blood pressure; DBP, diastolic blood pressure; CVD, cardiovascular disease.

[a]Central obesity defined as waist circumference $\geq$94 cm in males and $\geq$80 cm in females.
[b]High waist-to-hip ratio $\geq$0.95 for males and $\geq$0.80 for females.
[c]Education category "post-secondary" includes persons with vocational training, college, or university education; secondary corresponds to high school (up to grade 11); less than secondary corresponds to persons who had only elementary school education or persons who did not complete high school (i.e., high school grade 10 or below).

**Table 3  Odds ratio for elevated blood pressure or hypertension for putative risk factors among male and female participants in the 1986 Jamaica Birth Cohort.**

| Variable | Males $n = 409$ | | | Females $n = 489$ | | |
|---|---|---|---|---|---|---|
| | Odds ratio | 95% CI | P-value | Odds ratio | 95% CI | P-value |
| Age (years) | 1.48 | 1.03–2.12 | 0.034 | 2.16 | 1.41–3.31 | <0.001 |
| Height (cm) | 1.02 | 0.98–1.05 | 0.305 | 1.06 | 1.01–1.11 | 0.009 |
| BMI Category | | | | | | |
| *Normal weight* (18.5 –24.9 kg/m$^2$) | 1.0 | – | – | 1.0 | – | – |
| *Underweight* (<18.5 kg/m$^2$) | 0.52 | 0.17–1.58 | 0.250 | 1.31 | 0.61–2.83 | 0.489 |
| *Overweight* (25–29.9 kg/m $^2$) | 1.54 | 0.83–2.85 | 0.169 | 1.58 | 0.82–3.05 | 0.172 |
| *Obese* (≥30 kg/m$^2$) | 7.81 | 2.98–20.48 | <0.001 | 1.95 | 0.89-4.29 | 0.097 |
| Central obesity[a] | 6.57 | 2.48–17.36 | <0.001 | 1.54 | 0.88–2.72 | 0.132 |
| High glucose (upper quintile) | 2.14 | 1.35–3.39 | 0.001 | 1.20 | 0.48–3.01 | 0.693 |
| High cholesterol (≥5.2 mmol/l) | 1.77 | 0.88–3.56 | 0.110 | 1.62 | 0.88–2.97 | 0.120 |
| High LDL[b] (≥4.1 mmol/l) | 1.31 | 0.43–4.00 | 0.634 | 1.25 | 0.53–2.94 | 0.611 |
| Low HDL[c] | 1.11 | 0.67–1.83 | 0.694 | 1.24 | 0.71–2.17 | 0.449 |
| High triglycerides (upper quintile) | 1.80 | 1.08–2.99 | 0.024 | 1.96 | 1.10–3.51 | 0.023 |
| Creatinine (μmol/L) | 0.99 | 0.98–1.01 | 0.421 | 1.00 | 0.99–1.01 | 0.840 |
| HOMA-IR[d] (log, upper quintile) | 3.46 | 1.83–6.57 | <0.001 | 1.81 | 1.01–3.26 | 0.046 |
| White blood cell count | 1.02 | 0.89–1.17 | 0.781 | 1.12 | 0.99–1.27 | 0.084 |
| Albuminuria | 1.25 | 0.49–3.17 | 0.641 | 1.14 | 0.46–2.82 | 0.784 |
| High hsCRP[e] | 1.00 | 0.44–2.28 | 1.000 | 1.51 | 0.79–2.88 | 0.214 |
| Parental education | | | | | | |
| *Post–secondary* | 1.0 | – | – | 1.0 | – | – |
| *Secondary* | 0.92 | 0.54–1.59 | 0.777 | 1.29 | 0.68–2.46 | 0.433 |
| *Less than secondary* | 0.90 | 0.40–2.05 | 0.800 | 1.22 | 0.52–2.89 | 0.646 |
| Occupation of household head | | | | | | |
| *Professionals/Managers* | 1.0 | – | – | 1.0 | | |
| *Office, service or trade* | 0.74 | 0.43–1.28 | 0.279 | 1.36 | 0.65–2.83 | 0.416 |
| *Semi–skilled/Unskilled* | 0.89 | 0.48–1.63 | 0.702 | 1.83 | 0.95–4.50 | 0.067 |
| No. of household possession | | | | | | |
| *High* (15–17 items) | 1.0 | – | – | 1.0 | – | – |
| *Moderate* (10–14 items) | 0.58 | 0.33–1.02 | 0.061 | 4.36 | 1.31–14.51 | 0.016 |
| *Low* (0–9 items) | 0.81 | 0.43–1.53 | 0.515 | 2.76 | 0.79–9.72 | 0.113 |
| Physical activity level | | | | | | |
| *High* | 1.0 | – | – | 1.0 | – | – |
| *Moderate* | 0.83 | 0.52–1.32 | 0.425 | 0.82 | 0.38–1.75 | 0.605 |
| *Low* | 0.71 | 0.38–1.32 | 0.280 | 0.63 | 0.30–1.36 | 0.243 |
| Current cigarette smoking | 1.05 | 0.57–1.94 | 0.879 | 0.99 | 0.33–2.92 | 0.978 |
| Current marijuana smoking | 0.99 | 0.63–1.57 | 0.991 | 0.82 | 0.28–2.39 | 0.711 |
| Alcohol consumption | | | | | | |
| *Never drank alcohol* | 1.0 | – | – | 1.0 | —— | – |
| *Rarely drinks alcohol* | 1.01 | 0.39–2.67 | 0.978 | 0.60 | 0.29–1.24 | 0.166 |

**Table 3** (*continued*)

| Variable | Males $n = 409$ | | | Females $n = 489$ | | |
|---|---|---|---|---|---|---|
| | Odds ratio | 95% CI | *P*-value | Odds ratio | 95% CI | *P*-value |
| *Drinks 1–2 times/week* | 1.35 | 0.52–3.48 | 0.534 | 0.58 | 0.26–1.34 | 0.205 |
| *Drinks ≥3 times/week* | 1.16 | 0.45–2.94 | 0.762 | 0.47 | 0.19–1.14 | 0.095 |

**Notes.**
[a] Central obesity defined as waist circumference ≥ 94 cm in males and ≥80 cm in females.
[b] LDL, low density lipoprotein cholesterol. Estimates for high LDL did not include imputed values, due to high correlation with high cholesterol leading to potential problems with perfect prediction in imputation models.
[c] HDL, high density lipoprotein; Low HDL defined as <1.0 mmol/L for males and <1.3 mmol/L for females.
[d] HOMA-IR, Homeostasis Model Assessment–Insulin Resistance.
[e] hsCRP, high sensitivity C-reactive protein; absolute value for odds ratio for males = 0.9998.
We did not compute odds ratios for elevated waist-to-hip ratio given the very small number of males ($n = 4$) with high waist to hip ratio.

Results from the multivariable regression models are shown in Table 4. Models included variables as shown in the table and were done separately for males and females. Modifiable risk factors associated with higher odds of EBP/HTN among males were: obesity (OR 8.48, 95%CI [2.64–27.2], $p < 0.001$) and high glucose (OR 2.01, CI [1.20–3.37], $p = 0.008$). High HOMA-IR was also associated with higher odds of EBP/HTN among males, but did not achieve statistical significance (OR 2.08, CI [0.94–4.58], $p = 0.069$). Among females, modifiable risk factors associated with higher odds of EBP/HTN were: high triglycerides (OR 1.98, CI [1.03–3.81], $p = 0.040$), high HOMA-IR (OR 2.07, CI [1.03–4.12], $p = 0.039$) and lower SES (OR 4.63, CI [1.31–16.4], $p = 0.017$ [moderate vs. high household possessions]. OR for low vs. high household possessions was 2.61, CI [0.70–9.77], $p = 0.154$. The point estimates for obesity among females suggested higher odds of EBP/HTN, but this was not statistically significant (OR 1.44 CI [0.58–3.56], $p = 0.436$). Additionally, age was positively associated with EBP/HTN in both sexes and height in females only. Physical activity was inversely associated with EBP/HTN in both males and females with OR of 0.49 (CI [0.24–0.97]) and 0.42 (CI [0.18–0.97]) for low vs high physical activity level for males and females, respectively. Among women only, alcohol consumption was inversely related to EBP/HTN. Compared to those who never drank alcohol odds ratio for alcohol consumption <1 time per week was 0.41 (CI [0.18–0.90], $p = 0.026$), while for those report alcohol consumption ≥3 times per week odds ratio were 0.28 (CI [0.11–0.76] $p = 0.012$). Findings for the complete case analysis with models including 306 males and 409 females (Table S5) were generally similar to that obtained with multiple imputation, however estimates had wider confidence intervals and larger *p*-values, several of which did not achieve statistical significance.

We also ran multivariable ANOVA models with SBP and DBP as the outcome variables, in order to assess the robustness of our findings, and whether using high BP cut-points for classification may have influenced our results. These models are shown in Tables S8A and S8B. For the models with SBP as the outcome, the results for males were generally similar to the logistic regression model except that the association with physical activity level was no longer statistically significant and current cigarette smoking was now associated with higher odds of higher SBP. For females, obesity was now associated with higher systolic BP ($\beta$ 5.19 mmHg, CI [2.48–7.91], $p < 0.001$); household possessions was no longer retained in the model and the inverse association with alcohol consumption was now only significant

**Table 4  Factors associated with elevated blood pressure or hypertension (BP ≥120/80) in multivariable logistic regression models among male and female young adults in the Jamaica 1986 Birth Cohort.**

| Variable | Males ($n = 409$) | | | Females ($n = 489$) | | |
|---|---|---|---|---|---|---|
| | Odds ratio | 95% CI | P-value | Odds ratio | 95% CI | P-value |
| Age (years) | 1.74 | 1.16–2.61 | 0.007 | 2.55 | 1.60–4.08 | <0.001 |
| Height (cm) | – | – | – | 1.07 | 1.02–1.12 | 0.003 |
| BMI category | | | | | | |
| Normal weight (18.5–24.9 kg/m$^2$) | 1.0 | – | – | 1.0 | – | – |
| Underweight (<18.5 kg/m$^2$) | 0.64 | 0.20–2.00 | 0.441 | 1.70 | 0.74–3.91 | 0.211 |
| Overweight (25–29.9 kg/m$^2$) | 1.76 | 0.90–3.43 | 0.096 | 1.31 | 0.63–2.72 | 0.461 |
| Obese (≥30 kg/m$^2$) | 8.48 | 2.64–27.2 | <0.001 | 1.44 | 0.58–3.56 | 0.436 |
| High Glucose (upper quintile) | 2.01 | 1.20–3.37 | 0.008 | – | – | – |
| High Triglycerides (upper quintile) | – | – | – | 1.98 | 1.03–3.81 | 0.040 |
| HOMA-IR (log transformed, upper quintile) | 2.08 | 0.94–4.58 | 0.069 | 2.07 | 1.03–4.12 | 0.039 |
| High hsCRP | 0.45 | 0.17–1.17 | 0.101 | – | – | – |
| White blood cell count | – | – | | 1.14 | 0.99–1.31 | 0.076 |
| Household possessions | | | | | | |
| High (15–17 items) | 1.0 | – | – | 1.0 | – | – |
| Moderate (10–14 items) | 0.62 | 0.33–1.18 | 0.147 | 4.63 | 1.31–16.4 | 0.017 |
| Low (0–9 items) | 1.21 | 0.59–2.45 | 0.604 | 2.61 | 0.70–9.77 | 0.154 |
| Physical activity level | | | | | | |
| High physical activity level | 1.0 | – | – | 1.0 | – | – |
| Moderate physical activity level | 0.55 | 0.33–0.93 | 0.026 | 0.71 | 0.31–1.65 | 0.429 |
| Low physical activity level | 0.49 | 0.24–0.97 | 0.042 | 0.42 | 0.18–0.97 | 0.043 |
| Alcohol consumption | | | | | | |
| Never drank alcohol | – | - | – | 1.0 | – | – |
| Rarely drinks alcohol (<once/week) | – | – | – | 0.41 | 0.18–0.90 | 0.026 |
| Drinks alcohol 1–2 times/week | – | – | – | 0.46 | 0.19–1.15 | 0.099 |
| Drinks alcohol ≥3 times/week | | – | – | 0.28 | 0.11–0.76 | 0.012 |

**Notes.**
BMI, Body mass Index; HOMA-IR, Homeostasis Model Assessment Insulin Resistance; hsCRP, high sensitivity C-reactive protein.
Separate models created for males and females. Variables for inclusion in the final models were selected using backwards stepwise selection. Models for males included age, BMI category, high glucose, high HOMA-IR, High hsCRP, possession category and physical activity levels. Models for females included age, height BMI category, high triglycerides, high HOMA-IR, possession category, physical activity levels and alcohol consumption categories.

only for the 'rarely drinks alcohol category'. For models with DBP as the outcome, the only metabolic risk factor showing a significant association was high triglycerides, which was inversely associated among females only. Alcohol consumption was again inversely associated with DBP among females and was positively associated with DBP among males. Physical activity was again inversely associated with DBP among both males and females, but was statistically significant only for low vs high physical activity among females. Current marijuana use was inversely associated with DBP among males only.

An additional model with hypertension, defined using the 2017 ACC/AHA criteria, as the outcome was also obtained and shown Table S9. Significant correlates of hypertension in this model included high glucose among males and high triglycerides among females.

## DISCUSSION

In this study, we have found that the prevalence of EBP/HTN among young adults in Jamaica is higher among males compared to females and that there were significant sex differences in the relationship between EBP/HTN and some of the risk factors explored. EBP/HTN was positively associated with obesity and high glucose among males, and with high triglycerides, high HOMA-IR and fewer household possessions among females. Higher levels of HOMA-IR was also associated with EBP/HTN among males, but this did not achieve statistical significance. Physical activity was inversely associated with EBP/HTN in both males and females, while alcohol consumption was inversely associated with EBP/HTN in females only. The findings of this study are generally consistent with the published literature, but there are noteworthy findings as discussed below.

The overall prevalence of EBP/HTN among young adults in this study was lower than that reported for similar BP categories in a study from Uganda (*Kayima et al., 2015*), another in India (*Kini et al., 2016*) and among young adults in the Bogalusa Heart Study (*Toprak et al., 2009*), although the populations studied were generally older, with age ranging from 18–44 years. Kayima and colleagues *(2015)* reported a prehypertension prevalence of 40% and hypertension prevalence of 15% among young adults 18–40 years old in Uganda, while Kini and colleagues *(2016)* reported of prehypertension prevalence of 45% and hypertension prevalence of 3% among persons 20–30 years old in the Udupi District in India. In the Bogalusa Heart Study (*Toprak et al., 2009*), prehypertension prevalence was 37% and hypertension prevalence 13% among persons 20–44 years old. Prevalence of combined prehypertension or hypertension was also higher among indigenous youth in the USA (50% among persons 14–39 years old) and in Australia (55% among 15-24 year olds) (*Drukteinis et al., 2007*; *Esler et al., 2016*) Our prevalence estimates seem more aligned with the 12–17% prevalence of prehypertension among adolescents (mainly from the United States) quoted by Redwine and Daniels (*Redwine & Daniels, 2012*), but still lower than the 25% prevalence reported by Amma and colleagues *(2015)* from Kerala in India. We also note that the prevalence of EBP/HTN in this study was also lower than the 29% prehypertension prevalence reported for adolescents and young adults, 15–19 years old, from a national survey in Jamaica (*Ferguson et al., 2011b*) and the 31% prehypertension prevalence reported for adolescents and young adults 15–24 years old from a another national survey in Jamaica (*Wilks et al., 2008*). These findings suggest the prevalence of elevated blood pressure may be lower among young adults in urban Jamaica, and corroborates the finding of higher prevalence of hypertension among rural Jamaicans previously reported (*Ferguson et al., 2011a*). The findings also support the recent reports for higher prevalence of hypertension in Africa and Southeast Asia compared to the Americas (*Mills et al., 2016*; *NCD Risk Factor Collaboration, 2016*).

The finding of higher prevalence of EBP/HTN among young males has been a consistent finding for studies in Jamaica and elsewhere, including among indigenous youth in Australia and among American Indians (*Drukteinis et al., 2007*; *Esler et al., 2016*; *Ferguson et al., 2011b*; *Ferguson et al., 2008*; *Kayima et al., 2015*; *Kini et al., 2016*; *Toprak et al., 2009*). The mechanisms underlying this sex difference in EBP/HTN have not been clearly elucidated,

however Maranon and Reckelhoff suggested that both sex steroids and sex chromosomes effects, mediated by effects on the sympathetic nervous system, renin angiotensin system and sodium reabsorption contribute to the observed sex differences (*Maranon & Reckelhoff, 2013*). Joyner and colleagues (*(2016)* suggested that these sex differences may be mediated primarily through differences in $\beta$-adrenergic and $\alpha$-adrenergic mechanisms, which vary in relation to both age and sex.

We also found that while EBP/HTN was related to obesity the relationship was much stronger in males and, in fact, not statistically significant in females. We note that there were fewer females with EBP/HTN, thus analyses for females had less power to show a statistically significant effect. Additionally, when SBP was used as the outcome a statistically significant relationship was seen for both sexes. The magnitude of the effect in males was, however, much larger in males, with an eight-fold increase in the relative risk compared to a mere 50% increase in females, and regression coefficient in males being almost twice the value for females. The association between increased weight and BP is well established and is thought to be mediated through sympathetic nervous system activation, production of vasoactive adipocytokines and insulin resistance (*Acelajado, Calhoun & Oparil, 2013*; *Lloyd-Jones & Levy, 2013*). Our finding of a stronger effect of obesity on BP among males compared to females is supported by the findings of Cutler and colleagues *(2008)* where the increase in hypertension prevalence between the National Health and Nutrition Examination Survey (NHANES) 1988–1994 and NHANES 1999–2004 was almost completely explained after adjusting for BMI in males but not in females. The reasons for the larger effect of obesity on BP in men compared to women is not fully understood, however one study using 24-hour ambulatory BP monitoring found that average heart rate was higher in obese compared to non-obese men while heart rates were similar in both obese and non-obese women (*Kagan et al., 2006*). This suggest that sympathetic nervous system activation may be greater in obese men versus obese women and could be a possible mechanism for the sex-differences in the effect of obesity on BP. Our findings suggest that while reducing obesity is an important component of any public health intervention to reduce the prevalence of hypertension, the effect among women may be less than among men and therefore multi-component interventions may be more appropriate.

Insulin resistance is now generally accepted as a causative factor in hypertension (*Soleimani, 2015*; *Zhou, Wang & Yu, 2014*). Proposed mechanisms for hypertension in insulin resistance appear to be related primarily to renal handling of sodium, and possibly to vasodilatory effects, vascular smooth muscle cell proliferation and proinflammatory activity (*Nakamura et al., 2015*; *Soleimani, 2015*; *Zhou, Wang & Yu, 2014*). Higher levels of HOMA-IR was associated with EBP/HTN among females and borderline significant among males in this study. Additionally, high triglycerides was associated with EBP/HTN among women. HOMA-IR, as well as fasting insulin and triglycerides, are considered good surrogates measures on insulin resistance (*Abbasi, Okeke & Reaven, 2014*; *Matthews et al., 1985*; *McLaughlin et al., 2005*). While we acknowledge that there are some limitations in the use of HOMA in assessing insulin resistance (*Thompson et al., 2014*) as was done in this study, its strong correlation with clinical measures of insulin resistance suggest that this is a useful method for epidemiological studies (*Thompson et al., 2014*). Our findings suggest

that insulin resistance may be a potential target for preventing elevated BP. Potential approaches to reducing insulin resistance include weight reduction, as well as targeting renal sodium handling or improving insulin sensitivity (*Soleimani, 2015*; *Tsai et al., 2014*).

The association between lower socioeconomic status (based on number of household possessions) and elevated BP among females in this study adds to the growing body of literature on this subject. We have previously reported the lower maternal socioeconomic status (based on maternal occupation) was related to systolic blood pressure in a previous paper from this cohort, although in that report the effect was seen primarily among males (*Ferguson et al., 2015*). This sex differential in the social determinants of chronic diseases has been a consistent feature in studies from Jamaica (*Ferguson et al., 2010a*; *Ferguson et al., 2010b*; *Mendez et al., 2003*; *Mendez et al., 2004*) and suggests that sex differences must be considered when designing social interventions in similar populations.

While high levels of alcohol consumption is associated with hypertension, several studies have reported a J-shaped relationship, with light to moderate alcohol consumption being associated with reduced BP, particularly among women (*Cushman, 2001*; *Di Castelnuovo et al., 2006*; *Fisher, Orav & Chang, 2017*; *Fuchs et al., 1995*; *Sesso et al., 2008*). Our finding of an inverse relationship between alcohol consumption and EBP/HTN among women is therefore consistent with the literature, if we consider the highest drinking category of $\geq 3$ times/week as being consistent with moderate alcohol consumption. Unfortunately, we did not have data on the actual quantity of alcohol consumed and therefore were unable to explore this in the analysis.

The finding of lower odds of EBP/HTN among persons with lower physical activity was unexpected as a large body of literature supports an association between higher physical activity levels and lower BP (*Diaz & Shimbo, 2013*; *Whelton et al., 2002*). There are however a few studies where high physical activity was found to be associated with higher BP (*Teh et al., 2015*; *Tsioufis et al., 2011*). Possible explanations for these unexpected findings include misclassification of physical activity level, particularly in studies where physical activity level is based on self-report, residual confounding or a chance finding. Tsioufis and colleagues (*(2011)* suggest that increased stroke volume and stimulation of mechanically sensitive muscle receptors could be a possible explanation of their finding of higher BP among physically active adolescents in Greece. However, considering the large body of literature supporting the inverse relationship between physical activity and BP, chance, bias or residual confounding seem to be more plausible explanations.

This study had some limitations. Firstly, while the study had a moderately large sample size, the number of females with EBP/HTN was relatively small and therefore would result in low power to show statistically significant effects for some comparisons. Additionally, we did not have data on salt intake, which is generally considered an important determinant of blood pressure. The unavailability of data on quantity of alcohol consumed is also a limitation. We also acknowledge that the data used in this analysis are somewhat dated and although we do not expect that the biological effects of the putative risk factors would have changed since the data were collected, it is possible that the distribution of risk factors may have changed and as such the magnitude of some effects may also have changed. That a large proportion of participants had at least one missing value for variables of interest was

another limitation of this study. We were, however, able to overcome this by using multiple imputation methods for bivariate and multivariable analyses. Multiple imputation reduces the potential selection bias associated with complete case analysis and also improves the power of the study by facilitating use of data from incomplete cases (*Nguyen, Carlin & Lee, 2017*; *White, Royston & Wood, 2011*). Finally, the cross-sectional design precludes us making causal inferences from this study, however the risk factors studied have previously shown associations in longitudinal studies, suggesting that these may be true determinants of elevated blood pressure. Further studies with prospective evaluation of these risk factors in Afro-Caribbean populations would provide stronger evidence for causal associations.

Strengths of the study include the fact that we made detailed measurements of several CVD risk factors in an Afro-Caribbean population, thus providing data that will help reduce the dearth of data from the region. To our knowledge, this is the first study to evaluate effect of metabolic risk factors such and insulin resistance and triglycerides on elevated BP among Afro-Caribbean youth. The fact that we conducted sex specific analyses also allows for the exploration of separate approaches to interventions targeted at men and women. This study may help in informing future studies and in identifying potential targets for interventions aimed at reducing the adverse impact of elevated blood pressure in Afro-Caribbean populations. Additionally, this study could provide support for a public health policy targeting the reduction in metabolic risk factors in children and young adults to reduce the future burden of hypertension and other cardiovascular diseases.

## CONCLUSIONS

We have found that obesity, high glucose, high triglycerides and high HOMA-IR are correlates of EBP/HTN among Afro-Caribbean young adults in Jamaica and that there are significant sex differences in the relationship between some risk factors and EBP/HTN. Lower socioeconomic status was also positively associated with EBP/HTN in females, while alcohol consumption was inversely associated. These factors should be further explored in longitudinal studies, studies with larger sample sizes or meta-analyses and should be considered when designing interventions to ameliorate the adverse consequences of EBP/HTN. There is also a need for more mechanistic studies to understand the aetiology of EBP/HTN in Black populations.

## ACKNOWLEDGEMENTS

The authors acknowledge the contribution of the project staff (nurses, laboratory personnel, administrative staff, and project assistants) for their contribution to the project.

### Funding

This work was supported by research grants and staff support from the Caribbean Health Research Council (CHRC), the Caribbean Cardiac Society (CCS), the National Health Fund (NHF) Jamaica, the Culture, Health, Arts, Sports and Education (CHASE) Fund,

the University Hospital of the West Indies (UHWI) and the University of the West Indies, Mona Campus. The funders had no role in study design, data collection and analysis, decision to publish, or preparation of the manuscript.

### Grant Disclosures
The following grant information was disclosed by the authors:
Caribbean Health Research Council (CHRC).
Caribbean Cardiac Society (CCS).
National Health Fund (NHF).
Culture, Health, Arts, Sports and Education (CHASE).
University Hospital of the West Indies (UHWI).
The University of the West Indies, Mona Campus.

### Competing Interests
The authors declare there are no competing interests.

### Author Contributions
- Trevor S. Ferguson conceived and designed the experiments, performed the experiments, analyzed the data, wrote the paper, prepared figures and/or tables, reviewed drafts of the paper.
- Novie O.M. Younger-Coleman conceived and designed the experiments, performed the experiments, analyzed the data, contributed reagents/materials/analysis tools, reviewed drafts of the paper.
- Marshall K. Tulloch-Reid and Rainford J. Wilks conceived and designed the experiments, performed the experiments, contributed reagents/materials/analysis tools, reviewed drafts of the paper.
- Nadia R. Bennett reviewed drafts of the paper, contributed to the interpretation of data.
- Amanda E. Rousseau prepared figures and/or tables, reviewed drafts of the paper, contributed to the checking and interpretation of data.
- Jennifer M. Knight-Madden conceived and designed the experiments, performed the experiments, reviewed drafts of the paper.
- Maureen E. Samms-Vaughan and Deanna E. Ashley conceived and designed the experiments, reviewed drafts of the paper.

### Human Ethics
The following information was supplied relating to ethical approvals (i.e., approving body and any reference numbers):

The study was approved by the University of the West Indies/Faculty of Medical Sciences Ethics Committee.

### Data Availability
The raw data is provided in a Supplemental File.

## Supplemental Information

Supplemental information for this article can be found online at http://dx.doi.org/10.7717/peerj.4385#supplemental-information.

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
