# Peer review of "Factors associated with elevated blood pressure or hypertension in Afro-Caribbean youth: a cross-sectional study"

_PeerJ, doi:10.7717/peerj.4385_

## Round 0.1 · original submission · Major Revisions

I have read carefully the reports of the reviewers and your own paper. After this analysis, I think your manuscript should be re-evaluated assessing the indicated comments. Taking into account that these comments are so important, my decision is MAJOR REVISION.

With respect and warm regards,
Dr Palazón-Bru (academic editor for PeerJ)

My own comments:

1. The data source is old. This should be mentioned in limitations.
2. I would like to see more particulars about the education system in Jamaica. For example, I do not understand what a grade 11 is.
3. Please, explain the rationale of the groups performed in the quantitative variables.
4. No information is given about the sample size calculation. This is very important to know if your sample size is enough to answer your research question. Taking into account that you used a collected sample, this calculation should be performed a posteriori.
5. I guess you obtained standard deviations and absolute frequencies for the descriptive analysis, but they are not indicated.
6. How did you assess the goodness-of-fit of your regression models (discrimination and calibration)?

·

Basic reporting

No comment

Experimental design

No comment

Validity of the findings

No comment

Additional comments

The authors report an interesting cross-sectional study on factors associated with prehypertension in 898 young adults (mean age, 18.8 years). All subjects were born between September and October 1986, then similar in terms of age.
The study describes the burden of metabolic disorders such as obesity, high glucose and triglycerides in this large cohort of young Afro Caribbean adults. Authors show that metabolic disorders and socioeconomic status are associated with prehypertension a clear risk factor of hypertension development. Gender analysis is also interesting and discloses differences in prehypertension associated factors.
The study results may be able to support public health policy option to decrease the prevalence of metabolic disorders in young Afro Caribbean adults.

I have only minor concerns

1) In Table 3: I think that the 3 stars associated with “Binge Drinking” should be deleted.
2) In Table 4: The factor “Parental Occupation” may be deleted from the table since no association with prehypertension in multivariate analysis is shown.
3) I suggest that authors could propose that the results of their study may support a public health policy that focused on metabolic disorders decrease in childhood and in young adults.

Reviewer 2 ·

Basic reporting

This paper is presented in a clear, unambiguous, and professional manner, it conforms to professional standards of courtesy and expression. The authors have done a reasonable job and citing the most up to date references, although they have neglected to report on the CVD outcomes from the Modeling the Epidemiologic Study, a large cohort study currently following young adults (25-45yrs) in Jamaica, and 4 other African-origin adults e.g. Cardiovascular risk status of Afro-origin populations across the spectrum of economic development: findings from the Modeling the Epidemiologic Transition Study. Dugas LR, Forrester TE, Plange-Rhule J, Bovet P, Lambert EV, Durazo-Arvizu RA, Cao G, Cooper RS, Khatib R, Tonino L, Riesen W, Korte W, Kliethermes S, Luke A. BMC Public Health. 2017 May 12;17(1):438. And Elevated hypertension risk for African-origin populations in biracial societies: modeling the Epidemiologic Transition Study. Cooper RS, Forrester TE, Plage-Rhule J, Bovet P, Lambert EV, Dugas LR, Cargill KE, Durazo-Arvizu RA, Shoham DA, Tong L, Cao G, Luke A.J Hypertens. 2015 Mar;33(3):473-80; discussion 480-1. The current study should be discussed in relation to these findings.
Furthermore, many of the methods are not referenced (e.g. Physical activity questionnaire?) and some of the sentences are incomplete: e.g. sentence 88: where do estimates range from 2-19%, and were are estimates ranging from 12-45%?
The authors also present prevalence data for Jamaica: lines 107-111, however prevalence may be changing as a result of increased screening, it would be helpful to know incidence data. Perhaps increases are simply as a result of more people undergoing screening.
The authors are inconsistent throughout the manuscript with the use of abbreviations. Sometimes they use the abbreviation before the word is defined, other times they present the whole word, and then only abbreviate later.
This reviewer also feels that there are too many tables (especially supplemental tables), representing models that do not change the outcome of the paper, or fail to reach statistical significance.
The authors have submitted a paper which was rejected by another journal, and not removed the edits or formatting requested by that journal. It is recommended that the authors submit a clean version of this manuscript, fulling PEERJ’s formatting style.

Experimental design

While the overall aims, and research question are well presented and the paper does present with some novel findings there are several issues for this reviewer.
1) It is not clearly stated that participants were overnight fasted
2) How is the WHR defined?
3) HOMA-IR used fasted insulin and glucose, which is presented in the paper, why not calculate it?
4) No reference for the physical activity questionnaire
5) No references for SES questionnaires
6) The authors imputed biochemical data, this is a serious weakness in the current data. There is no data presented to show how the associations changed with or without the imputed data? Also up to a quarter of some of the variables were missing: Family history of HTN. This would seem to be a major flaw if HTN data is missing?
7) The authors did not adjust their models for smoking/ganja, this seems as a major flaw for predicting future risk for HTN.

Validity of the findings

This paper addresses a unique population for early diagnoses of HTN among young adults. However, this reviewer has several issues with the statistical analyses (lack of adjusting for smoking), and imputation of missing variables (how do outcomes change with or without imputation?). The paper also does not include several references, see basic reporting section. Throughout the paper, it is not clearly stated who comparisons are being compared to, e.g line 326-327. This reviewer suggests cutting down on the tables, particularly those representing models that fail to reach statistical significance. The authors have complied with the data sharing policy.

Additional comments

Overall this is a well written paper presenting novel findings in an under-studied age group. This reviewer suggests that the authors be given the opportunity to address the comments, and particularly the analyses and then resubmit their work.

Reviewer 3 ·

Basic reporting

No comment

Experimental design

The study is a retrospective case control study with cases defined as Afro-Caribeean youth having elevated blood pressure (either pre-hypertension or hypertension) The reviewer congratulates the research team in recognising the important contribution that pre-hypertension plays as a cardiovascular risk factor The study seeks to fill an important evidence gap pertaining to elevated blood pressure in youth from a specific ethnic group.
My concern with the study design is the lack of a subset analysis looking at hypertension (preferably stratified into Type I and Type II hypertension). The reason for this is two fold. Firstly, from a public health perspective understanding the risk factors for youth with established hypertension as opposed to just pre-hypertension is important as it assists in prioritising preventive action. The other reason is that readers less familiar with the concept of pre-hypertension may be misled by the combination of cases being defined as elevated hypertension. By reporting the results categorically, and combined it will better inform readers and inform practice. I appreciate that the prevalence of hypertension is low which may preclude this sort of analysis however this should be mentioned explicitly if the authors believe this is the cas.e
The association between increased ACR and hypertension is not mentioned in the results - despite this data being available. I would like to see this finding mentioned. As an early marker of renal impairment it would be interesting to know if it contributes to elevated BP in this group.
The research question is well defined and appropriate methodology and statistical analysis is used.

(5) Ethical clearance for the article was evidenced.

Validity of the findings

(1) In the discussion I would like to see further comparison with other studies of elevated BP in youth. Eg the gender difference that was found here has been found among Australian Indigenous youth.
(2) The limitations of the study are appropriately delineated.
(3) Further exploration of the public health implications of the study would be beneficial

---

## Round 0.2 · accepted · Accept

Dear authors,

I am happy to report that your paper has high standards to be published in PeerJ in its current form.

Congratulations!

With respect and warm regards,
Dr Palazón-Bru (academic editor for PeerJ)

·

Basic reporting

no comment

Experimental design

no comment

Validity of the findings

no comment

Additional comments

Author's answers are appropriate with my requests.
I have no more comments
Thank You